# Cholesterol and SREBP2 Dynamics During Spermatogenesis Stages in Rabbits: Effects of High-Fat Diet and Protective Role of Extra Virgin Olive Oil

**DOI:** 10.3390/ijms26094062

**Published:** 2025-04-25

**Authors:** María Virginia Avena, Abi Karenina Funes, María Ángeles Monclus, Paola Vanina Boarelli, Luis Fernando Barbisan, M. Rosa Bernal-López, Ricardo Gómez-Huelgas, Tania Estefania Saez Lancellotti, Miguel Walter Fornés

**Affiliations:** 1Andrological Research Laboratory of Mendoza (LIAM), IHEM, National University of Cuyo, CONICET, Mendoza M5502JMA, Argentina; virgiavena@gmail.com (M.V.A.); abikarenina2020@gmail.com (A.K.F.); marianmonclus@gmail.com (M.Á.M.); 2Metabolic Diseases Laboratory (LEM), Juan Agustín Maza University, Mendoza M5519, Argentina; pboarelli@umaza.edu.ar; 3Biosciences Institute, Department of Structural and Functional Biology, São Paulo State University (UNESP), Botucatu 18610-034, SP, Brazil; luis.barbisan@unesp.br; 4Internal Medicine Department, Regional University Hospital of Málaga, Málaga Biomedical Research Institute (IBIMA), 29009 Málaga, Spain; rosa.bernal@ibima.eu (M.R.B.-L.); rgh@uma.es (R.G.-H.); 5Biomedical Research Networking Center on Obesity and Nutrition (CIBERobn), Carlos III Health Institute, 28029 Madrid, Spain

**Keywords:** cholesterol, SREBP2, stages of the seminiferous epithelium cycle, spermatogenesis, rabbits, high-fat diet, extra virgin olive oil (EVOO)

## Abstract

High-fat diets (HFDs) have been found to compromise male fertility, with cholesterol dysregulation being a key factor. Sterol regulatory element-binding protein 2 (SREBP2) is a crucial transcription factor that regulates cholesterol biosynthesis and uptake, playing an essential role in maintaining cholesterol homeostasis in the testes. This study investigated the dynamics of SREBP2 and cholesterol levels during rabbit spermatogenesis under HFD conditions. Our findings reveal that SREBP2 expression fluctuates throughout the seminiferous epithelium cycle. However, HFDs induce stage-specific disruptions in cholesterol balance, leading to sperm with increased membrane cholesterol, a reduced sperm count in semen analysis, impaired motility, abnormal morphology, and decreased functionality. In the control group, SREBP2 expression patterns underscored its critical role in normal spermatogenesis. Interestingly, supplementation with extra virgin olive oil (EVOO) reversed the negative effects of HFD, normalizing SREBP2 expression and cholesterol content, which improved sperm quality. These findings emphasize the importance of stage-specific analysis in understanding how dietary fat impacts male fertility and suggest that EVOO may serve as a potential nutritional intervention to protect reproductive health.

## 1. Introduction

The intricate relationship between diet and male reproductive health is a burgeoning area of investigation within biomedical research. HFDs have been consistently implicated in a decline in seminal quality, adversely affecting parameters such as sperm motility, viability, and morphology [1,2,3,4,5,6,7]. Our findings demonstrate a significant positive correlation between elevated serum cholesterol levels and increased sperm cholesterol content in a hypercholesterolemic rabbit model [8]. This cholesterol accumulation in sperm cells likely results from dysregulated cholesterol metabolism during spermatogenesis [9]. However, the specific molecular mechanisms underlying these detrimental effects remain elusive.

To investigate the impact of dietary cholesterol on male reproductive health, animal models of diet-induced hypercholesterolemia have proven to be valuable tools. Among these, the rabbit emerges as a particularly relevant translational model due to its metabolic similarities to humans, notably in cholesterol management and lipoprotein composition [10,11]. This approach allows a direct examination of how metabolic alterations, such as elevated serum cholesterol levels, impact critical processes during spermatogenesis.

One key regulator of intracellular cholesterol homeostasis is SREBP2 (sterol response element-binding proteins 2). This transcription factor, belonging to a family that includes SREBP1a, SREBP1c, and SREBP2, plays a pivotal role in cholesterol intracellular balance [12,13]. SREBP2, synthesized as a precursor protein (125 kDa), is anchored to the endoplasmic reticulum. Upon activation, it is transported to the Golgi apparatus by SCAP (sterol cleavage-activating protein). Subsequent proteolytic cleavage by site 1 and site 2 proteases (SP1 and SP2) releases the soluble active transcription factor, which translocates to the nucleus and binds to sterol response elements (SRE) in the promoters of genes involved in cholesterol synthesis, transportation, and uptake. Notably, sterols exert a negative feedback loop on this pathway by inhibiting SREBP through interactions with SCAP [14,15].

The seminiferous epithelium is fundamental to spermatogenesis, the complex process by which sperm cells are produced. This progression occurs in a highly organized manner, with spermatogenic cells transitioning through distinct stages of the seminiferous epithelial cycle. Each stage is characterized by specific cell associations and tightly regulated morphological and functional changes. These transitions demand significant metabolic adaptations, including precise regulation of cholesterol, a key molecule for maintaining membrane fluidity and cellular integrity [16,17,18,19]. Notably, elevated cholesterol levels have been linked to increased cholesterol accumulation in sperm cells [8], suggesting an upregulation of the intracellular cholesterol pathway as a compensatory response [9]. However, these alterations may not necessarily disrupt the overall structure of the seminiferous epithelium, as evidenced by the absence of significant histological abnormalities despite observed declines in sperm count.

Interestingly, it has been shown that supplementing a high-fat diet with extra virgin olive oil improves serum cholesterol levels, normalizes SREBP2 expression, and improves seminal parameters [20,21].

This study focuses on investigating the distribution of SREBP2 and cholesterol within the seminiferous epithelium, as they are key regulators of testicular cholesterol homeostasis. By examining the expression and regulation of SREBP2 across various stages of the seminiferous epithelial cycle, we aim to understand how dietary fats—specifically those rich in saturated and unsaturated fats—affect cholesterol balance at a molecular level. Our analysis also includes the effects of EVOO supplementation in restoring the balance of cholesterol and SREBP2 expression in this context.

## 2. Results

### 2.1. General Parameters

#### 2.1.1. Body Parameters

After 12 months under HFD, biometric measurements, including body weight, neck circumference (cm), and abdomen circumference (cm), did not show any differences compared to control diets (CD). Additionally, the body mass index (BMI), a critical indicator of nutritional status, remained comparable across experimental groups [20].

#### 2.1.2. Serum Analyses

No significant differences in glucose levels were observed between the groups, indicating that the experimental diets did not significantly impact glucose regulation. Regarding cholesterol, the HFD significantly increased total serum cholesterol levels compared to the control group (HFD: 87.02 ± 9.86 mg/dL; CD: 38.51 ± 7.6 mg/dL). However, a reduction in dietary fat lowered cholesterol levels (½ HFD: 43.91 ± 7.16 mg/dL), but did not reach control values. Supplementation with EVOO in the ½ HFD significantly reduced cholesterol levels, approaching control values (½ HFD + ½ EVOO: 41.53 ± 5.72 mg/dL). In rabbits fed exclusively with EVOO, cholesterol levels were similar to those of the protected and control groups, with no significant differences (EVOO: 47.95 ± 0.92 mg/dL). Concerning non-HDL cholesterol, the HFD markedly increased this parameter. Conversely, EVOO supplementation improved the lipid profile by increasing HDL cholesterol levels and reducing non-HDL cholesterol [9,21]. Finally, the analysis of liver enzymes (GOT and GPT) and other indicators of organ damage showed some variability between groups. However, no significant differences were observed, suggesting that the studied diets did not cause liver damage, as detailed in the previous study [20].

#### 2.1.3. Semen Analysis

High-fat diets (HFD and ½ HFD) negatively impacted seminal quality, as evidenced by a significant reduction in seminal volume, sperm concentration, and motility, along with increased morphological abnormalities. In contrast, diets supplemented with olive oil (EVOO and ½ HFD + ½ EVOO) maintained seminal parameters comparable to the control group, aligning with findings from previous studies [8,22].

### 2.2. Characterization of the Seminiferous Epithelium

#### 2.2.1. Testis Morphology—Stage Classification of Seminiferous Epithelium

The seminiferous epithelium within the seminiferous tubules undergoes a continuous cycle, characterized by the dynamic association of different spermatogenic cell types. This specific arrangement of cells defines the stages of the spermatogenic cycle. In rabbits, eight distinct stages of the cycle have been described by Swierstra et al., 1963 [23]. Figure 1 illustrates these stages, demonstrating the characteristic cellular associations within each stage. This figure could be useful for understanding the changes observed in this paper.

#### 2.2.2. Isolated Seminiferous Tubule Characterization and Correlation with Stages

To facilitate the study of cellular processes within specific stages of the spermatogenic cycle, we employed a method for isolating seminiferous tubules described by Mäkelä et al., which is based on tubule translucency [24]. This approach enables the classification of tubules into three distinct zones: Zone 1 (Z1): light zone, corresponding to stages I and II; Zone 2 (Z2): intermediate zone, corresponding to stages III, IV, V, and VI; and Zone 3 (Z3): dark zone, corresponding to stages VII and VIII. The validity of this three-zone classification was confirmed through detailed histological analysis (Figure 2).

This initial step enabled us to simultaneously obtain a large amount of tissue in similar stages of the seminiferous epithelial cycle, ensuring its suitability for subsequent experiments.

### 2.3. Cholesterol Analyses

#### 2.3.1. Cholesterol Accumulation in the Seminiferous Epithelium

Filipin staining revealed a significant increase in cholesterol accumulation within the seminiferous epithelium of HFD and ½ HFD animals compared to the control group, with higher levels observed in the HFD group than in the ½ HFD (Figure 3d–f,j–l). This accumulation was particularly evident in the apical region of the seminiferous epithelium, which contains elongated spermatids, cytoplasmic droplets, and spermatozoa close to spermiation. However, the incorporation of EVOO into the ½ HFD significantly attenuated this cholesterol accumulation (Figure 3m–o). Densitometric analysis of filipin staining further confirmed that EVOO supplementation significantly reduced testicular cholesterol levels (Figure 4). No significant differences were observed between the CD, the EVOO groups, and the ½ HFD group.

The results presented here, using the filipin cholesterol detection method, indicate that HFD induces cholesterol overload in the seminiferous tubules. However, EVOO supplementation significantly alleviated this excessive accumulation, particularly in the ½ HFD group.

#### 2.3.2. Cholesterol Distribution and Quantification Across Spermatogenic Stages

To further investigate the impact of dietary cholesterol on the seminiferous epithelial cycle, we examined cholesterol distribution across the different stages of the seminiferous epithelium. Using the three-zone classification based on tubule translucency (Z1, Z2, and Z3), we observed a progressive increase in cholesterol accumulation from Z1 to Z3 across all groups, except in the ½ HFD + ½ EVOO group, where cholesterol levels remained relatively stable across all zones (Figure 5 and Figure 6). This accumulation was most pronounced in the HFD group.

These results showed that cholesterol levels increased progressively from Z1 to Z3 under all conditions. The increase was particularly pronounced in HFD and ½ HFD groups, with the highest accumulation observed in HFD. However, the presence of EVOO in the diet significantly mitigated this rise. To better illustrate these differences between dietary groups, data were analyzed and represented according to diet and zone.

#### 2.3.3. Cholesterol Distribution by Diets and Zones

Individual analysis of each diet (Figure 7) revealed a gradual increase in cholesterol accumulation from Zone 1 to Zone 3 across all experimental groups, including the control group. Notably, even under basal conditions, Zone 3 exhibited a significant increase compared to Z1 and Z2. In the HFD and ½ HFD groups, cholesterol accumulation was evident from the start, with a significant increase across all three zones. In contrast, EVOO supplementation attenuated this rise, resulting in a substantial but less pronounced increase compared to diets without added fat. Finally, the ½ HFD + ½ EVOO group showed no cholesterol accumulation differences between zones.

Considering all filipin studies, cholesterol levels in seminiferous tubules are influenced by both diet and stage. High-fat diets induce a significant increase, while EVOO supplementation helps reduce cholesterol accumulation. This dietary impact alters the natural cholesterol wave observed under basal conditions, which is characterized by lower levels in Z1 and Z2 and a peak in Z3.

### 2.4. SREBP2 in Seminiferous Tubules

#### 2.4.1. SREBP2 Localization

Immunofluorescence analysis revealed an increased expression of SREBP2 within the seminiferous epithelium of HFD-fed animals compared to controls. This upregulation was particularly evident in both the basal and upper strata of the seminiferous epithelium (Figure 8a,d,g,j,m). In contrast, animals receiving a diet containing half HFD and EVOO showed a significant reduction in SREBP2 expression, with the most pronounced decrease observed when EVOO was included in the diet (Figure 8m–o). Densitometric analysis further confirmed these findings (Figure 9).

Positive immunostaining of SREBP2 reveals its presence, distribution, and relative abundance in the seminiferous epithelium under different dietary conditions.

#### 2.4.2. Distribution of SREBP2 Across Spermatogenic Stages

To investigate SREBP2 expression and localization, immunofluorescence staining was performed on different zones of the seminiferous epithelium, identified by transillumination, for each experimental group (Figure 10). SREBP2 expression was notably higher in stages enriched with round and elongated spermatids (Zone Z2, Figure 10b,e,h,k,o—white circles), and in the apical region of the tubules in more advanced stages (Zone Z3, Figure 10c,f,i,m,p—white squares).

Visual inspection suggested variations in SREBP2 signal intensity across zones (Z1–Z3) and between dietary groups (Figure 10). To quantify these observations, SREBP2 fluorescence intensity was measured within different epithelial strata across all groups.

Quantitative analysis revealed that the CD group exhibited a relatively consistent fluorescence signal across all zones (Figure 11). The HFD and ½ HFD groups showed a generalized increase in SREBP2 compared to CD, but with a gradual decrease in signal intensity from Z1 to Z3 (Figure 11). Conversely, the addition of olive oil to the ½ HFD significantly reduced the SREBP2 signal (row 5, Figure 10), with no significant differences observed across zones (Figure 11).

Figure 11 summarizes the quantitative analysis of SREBP2 levels in different zones of the seminiferous tubules according to dietary treatment.

#### 2.4.3. Distribution of SREBP2 Among Zones/Stages and Diet

Analysis of SREBP2 expression within each dietary group revealed distinct distribution patterns across the different zones/stages. In the control, EVOO, and HFD groups, a general trend of decreasing SREBP2 expression from Zone 1 to Zone 3 was observed. In contrast, the ½ HFD group exhibited an increase in SREBP2 expression towards Z3, while the ½ HFD + ½ AOVE group displayed a unique profile, with peak expression in Zone 2. Notably, the HFD group, despite demonstrating higher initial expression levels, also showed a consistent downward trend towards Z3 (Figure 12). Collectively, these results confirm the presence and quantify the differential expression of SREBP2 across dietary groups and zones/stages.

### 2.5. Comparison Between SREBP2 and Cholesterol Distribution in Seminiferous Tubules

#### Relationship Between SREBP2 and Cholesterol Distribution in the Seminiferous Epithelium

To investigate the interplay between SREBP2 and cholesterol regulation in the seminiferous epithelium, we compared their fluorescence intensities across three representative dietary groups (Figure 13): control (CD), high-fat diet (HFD), and a mixed diet (½ HFD + ½ EVOO, protective diet). Fluorescence intensity was represented by a line from very low signal to very high from Z1 to Z3. This comparative analysis revealed contrasting patterns in the distribution of these molecules across the seminiferous epithelium. While cholesterol fluorescence intensity increased progressively from Z1 to Z3, SREBP2 fluorescence exhibited a linear downward trend between Z1 to Z3. In CD and ½ HFD + ½ EVOO, both lines were located between very low to moderate, the lowest sector of the graph. In contrast, under HFD, the lines moved to the top of the graphics, with a high and very high signal. These opposing trends intersected at Z2 in CD and HFD groups, but this intersection was delayed, occurring between Z2 and Z3, in the ½ HFD + ½ EVOO group. Notably, fluorescence intensity, reflecting relative molecule concentrations, was higher in HFD groups compared to both CD and ½ HFD + ½ EVOO groups.

A comparison of fluorescence intensity between cholesterol and SREBP2 showed an increased signal from Z1 to Z3 or a linear downward slope, respectively, in any diets probed here. However, the threshold is higher for FHD than CD or ½ HFD + ½ EVOO (protective diet).

### 2.6. Molecular Studies

#### Expression of SREBP2 mRNA in the Seminiferous Epithelium

Animals fed with HFDs (HFD and ½ HFD) exhibited a significant increase in SREBP2 mRNA expression in testicular tissue compared to the control group (CD, Figure 14). This finding suggests enhanced activation of the cholesterol biosynthesis pathway in response to the high-fat diet. On the other hand, supplementation with extra virgin olive oil (½ HFD + ½ EVOO, protective diet) significantly attenuated this increase in SREBP2 mRNA expression. Notably, exclusive supplementation with extra virgin olive oil did not significantly alter SREBP2 mRNA expression levels compared to the control group.

Increases in SREBP2 mRNA expression were coincident with increases in immune detection of SREBP2, indicating an enhanced activation of the cholesterol biosynthesis pathway in response to the high-fat diet. On the other hand, supplementation with extra virgin olive oil (½ HFD + ½ EVOO, protective diet) significantly attenuated this behavior. Notably, exclusive supplementation with extra virgin olive oil did not significantly alter SREBP2 mRNA expression levels compared to the control group.

## 3. Discussion

Our findings provide new insights into the molecular mechanisms by which HFDs impair male fertility. We demonstrate that HFDs induce stage-specific alterations in SREBP2 expression and cholesterol distribution within the seminiferous epithelium, particularly affecting stages associated with sperm differentiation and spermiogenesis (Zones 2 and 3; stages IV–VIII of the seminiferous epithelial cycle). These disruptions are accompanied by abnormal sperm and seminal parameters, underscoring the detrimental effects of dietary lipids on spermatogenesis.

Numerous studies have demonstrated a close relationship between HFDs and altered spermatogenesis [25,26,27,28]. However, the negative impact of hypercholesterolemia on semen quality is not yet fully understood at the molecular level. This study reveals that intracellular cholesterol regulation, closely linked to circulating cholesterol, depends on the SREBP2-mediated pathway, a central regulator of lipid homeostasis [29].

Our study demonstrates that a HFD leads to significant alterations in serum cholesterol levels and seminal parameters, despite no notable differences in biometric measurements or BMI after 12 months of dietary exposure. These findings align with the concept of metabolic obesity with normal body weight, a condition characterized by systemic metabolic disruptions without overweight gain [20,30,31]. While glucose regulation remained unaffected, HFDs markedly increased total and non-HDL cholesterol levels. Conversely, supplementation with EVOO improved the lipid profile by reducing non-HDL cholesterol and increasing HDL cholesterol, returning values close to controls. Importantly, these dietary conditions did not induce hepatic damage, as evidenced by unchanged liver enzyme levels (GOT and GPT), corroborating previous findings on EVOO’s protective effects against dyslipidemia [20,32,33].

These dietary modifications had profound effects on the seminiferous epithelium and seminal parameters. Rabbits exposed to HFDs (HFD and ½ HFD) exhibited significant impairments in seminal quality, including reduced seminal volume, sperm concentration, and motility, accompanied by an increase in morphological abnormalities [8]. In stark contrast, EVOO-supplemented diets (EVOO and ½ HFD + ½ EVOO, protective diet) preserved seminal quality, with parameters comparable to the control group. These findings highlight the protective role of EVOO in mitigating the adverse effects of HFDs on male fertility, in agreement with previously reported evidence of its beneficial impact on lipid metabolism and reproductive health [9].

Sperm cells that migrate from the testis to the epididymis are the end product of a highly organized process within the “seminiferous epithelium factory”. This epithelium produces spermatozoa through a series of well-defined steps, where cells transition from the basal to the apical layer in a synchronized wave-like cycle known as the stages of the seminiferous epithelium [34]. During these stages, critical cellular processes such as meiosis and spermiogenesis occur. These processes are tightly regulated by various factors, including hormones, genes, and transcription factors [35]. Notably, cholesterol metabolism plays a crucial role in this regulation, with its distribution and pathways being closely linked to specific stages of the seminiferous epithelium [36]. By employing innovative histological and transillumination techniques, our study successfully isolated and analyzed distinct stages of the seminiferous epithelium, offering a detailed understanding of the molecular and metabolic processes underpinning spermatogenesis.

Under control conditions, a progressive increase in cholesterol accumulation was observed from zone 1 to zone 3 of the seminiferous epithelium, as evidenced by filipin staining. This pattern aligns with the progression of spermatogenesis, particularly from spermatogonia at the basal compartment to mature spermatozoa near the lumen, reflecting the metabolic and structural demands associated with cellular differentiation. This finding, representing the first detailed analysis of zonal cholesterol distribution during spermatogenesis, highlights the dynamic nature of cholesterol metabolism throughout sperm development.

High-fat diets significantly disrupted this physiological pattern, leading to an exaggerated accumulation of cholesterol, particularly in Zones 2 and 3, corresponding to stages III-VIII of the epithelial cycle, crucial for spermatid differentiation. This excessive cholesterol accumulation likely impairs membrane remodeling processes essential for sperm maturation, potentially contributing to the observed decline in semen quality [1]. Notably, supplementation with EVOO significantly mitigated this effect, restoring a more physiological pattern of cholesterol distribution within the seminiferous epithelium. These results highlight the protective role of EVOO against the lipotoxic effects of HFDs by preserving cholesterol homeostasis within the seminiferous epithelium and maintaining an optimal environment for spermatogenesis.

To investigate the molecular mechanisms underlying these observations, we examined the expression and localization of SREBP2, a key regulator of cholesterol homeostasis in the testis. SREBP2 is expressed in both Leydig cells and seminiferous tubules [5,16,19,37,38], playing a crucial role in cholesterol biosynthesis and cellular lipid metabolism. While previous studies suggested that cholesterol biosynthesis during spermatogenesis might be regulated by SREBP-independent mechanisms [39], the presence of isoforms such as SREBPgc, along with findings from the present work, indicate a complex regulatory system [16,17,19,40]. A recent proteomic study analyzed the mice testis fed HFDs and found changes in the expression of proteins involved in lipid homeostasis, including SREBP2 [6], highlighting the potential impact of dietary factors on SREBP2 activity in the testis. However, the specific role of SREBP2 in regulating cholesterol homeostasis within the seminiferous epithelium and its response to dietary perturbations remained largely unexplored.

Under control conditions, we observed an inverse relationship between cholesterol accumulation and SREBP2 expression across the seminiferous epithelium. As cholesterol levels increased from the basal to the apical regions, reflecting the increasing metabolic demands of spermatogenesis, SREBP2 expression progressively decreased. This suggests that SREBP2 activity is tightly regulated to prevent excessive cholesterol accumulation by negative feedback mechanisms, which may involve post-translational modifications, protein degradation, or transcriptional repression [41]. Cholesterol likely accumulates in cellular membranes and cytoplasmic droplets during spermatogenesis, while SREBP2 activity is gradually downregulated, potentially through mechanisms such as protein degradation, ubiquitination, or proteasomal degradation [42].

In contrast, HFDs significantly upregulated SREBP2 expression, particularly in the nuclei of spermatogenic cells [21], indicating chronic activation of the cholesterol biosynthetic pathway. This chronic SREBP2 activation likely drives excessive cholesterol accumulation, contributing to testicular lipotoxicity, a phenomenon previously observed with SREBP-1c in other tissues [43,44]. Additionally, lipophagy, a process regulating lipid droplet mobilization via autophagy, may be altered in this context, explaining some observed consequences [45]. Similar phenomena have been previously reported in other animal models and different durations of fat consumption [26,27,28]. In previous studies by our research group, the impact of HFDs during acute (6 months) and chronic (12 months) periods was analyzed, revealing increased cholesterol content in both cases [21]. However, acute cholesterol accumulation appears to be directly influenced by diet, whereas chronic accumulation is associated with pathway dysregulation. In animals consuming HFDs for less than six months, significant decreases in key pathway proteins (SREBP2, HMG-CoA, LDL-R) were observed compared to controls. However, after 12 months of HFD exposure, notable differences emerged, suggesting that the regulatory system fails to adequately detect high circulating lipid concentrations. As a result, the machinery for cholesterol synthesis and incorporation into cells is activated, further worsening the lipid imbalance.

EVOO supplementation effectively prevented the HFD-induced upregulation of SREBP2, restoring its expression levels and localization to control levels. This finding aligns with the known mechanisms of EVOO, which positively influence cholesterol metabolism by upregulating LDL receptors and inhibiting HMG-CoA reductase, key enzymes involved in cholesterol synthesis and uptake [46]. Given that high-fat diets contribute to elevated serum and testicular cholesterol levels, and have been shown to impair spermatogenesis [8,9], the cholesterol-lowering properties of EVOO are particularly relevant in this context. Previous studies have demonstrated that EVOO supplementation can counteract the detrimental effects of high-fat diets on testicular cholesterol levels, restoring key regulatory proteins and improving sperm parameters [21,22].

The observed inverse relationship between cholesterol and SREBP2 fluorescence signals provides valuable insights into their dynamic interplay. Under control conditions, the intersection of these signals occurs in Zone 2 of the seminiferous epithelium, reflecting a balanced regulatory system. In HFD-fed animals, this intersection point remains in Zone 2; however, increased cholesterol accumulation likely leads to altered SREBP2 activity, potentially through mechanisms such as increased proteolytic degradation (activation) or feedback inhibition. This disruption in the coordinated regulation of cholesterol and SREBP2 contributes to impaired spermatogenesis. Interestingly, EVOO supplementation effectively mitigated this disruption, delaying the intersection point towards later stages (between Z2 and Z3). This suggests that EVOO not only reduces cholesterol accumulation, but also modulates SREBP2 activity, potentially by downregulating its expression or activity, restoring a more balanced regulatory system within the seminiferous epithelium.

Furthermore, our findings demonstrate a significant increase in SREBP2 mRNA expression in animals fed HFDs compared to the control group, indicating enhanced activation of the cholesterol biosynthesis pathway. This increase was significantly attenuated by EVOO supplementation, further supporting its beneficial effects on lipid homeostasis within the testis. Notably, exclusive EVOO supplementation did not significantly alter SREBP2 mRNA expression levels compared to the control group.

These findings highlight the critical role of SREBP2 in regulating cholesterol homeostasis within the seminiferous epithelium and its sensitivity to dietary perturbations. In animals fed HFDs, we observed a chronic activation of the SREBP2 pathway, potentially contributing to the observed lipotoxicity and impaired spermatogenesis. This chronic activation likely involves a complex interplay of factors, including increased SREBP2 protein synthesis, altered post-translational modifications, and impaired protein degradation.

Chronic activation of the SREBP2 pathway under HFD conditions likely contributes to testicular lipotoxicity and impaired spermatogenesis. The observed alterations in SREBP2 expression and cholesterol distribution highlight a potential link between dietary lipids and male reproductive health, emphasizing the need for dietary interventions to mitigate these effects. The protective effects of EVOO, as demonstrated by improved lipid profiles, restored SREBP2 activity, and preserved seminal parameters, underscore its potential as a dietary strategy to counteract the detrimental effects of HFDs. Further studies are warranted to elucidate the molecular mechanisms underlying these protective effects and to explore their clinical implications for human health.

## 4. Materials and Methods

### 4.1. Reagents

Phosphate-buffered saline tablets (Sigma P4417, Merck, Darmstadt, Germany) were prepared by dissolving one tablet in 200 mL of distilled water (final concentration: 0.01 M, phosphate buffer, 0.027 M KCl, 0.137 M NaCl, pH 7.4). Determinations of glucose, triglycerides, total cholesterol (TC), and HDL cholesterol (C-HDL) was performed using GTLab specific kits (Rosario-Argentina), https://www.gtlab.com.ar/, accessed on 15 April 2025.

### 4.2. EVOO Analysis

Extra virgin olive oil composition: quality of olive oil as extra virgin was certified by an olive oil tasting panel (Panel de Cata, School of Agronomy at the National University of Cuyo, https://fca.uncuyo.edu.ar/categorias/index/panel-de-cata-mendoza-de-aceite-de-oliva, accessed on 15 April 2025).

Biochemical components were determined at the INTI (National Institute for Industrial Technology—Instituto Nacional de Tecnología Industrial, https://www.argentina.gob.ar/inti, accessed on 15 April 2025).

The corresponding tables about the quality of EVOO used are shown in the Appendix A.

### 4.3. Animal Model and Experimental Groups

The research group developed a translational medicine animal model (hypercholesterolemic New Zealand white rabbits) to study the association between high-fat diets and male infertility.

The work protocol was supervised and approved by the Institutional Committee for the Care and Use of Laboratory Animals (CICUAL http://fcm.uncuyo.edu.ar/paginas/index/cicual; 06_150702, accessed on 15 April 2025). Adult males (2–20 months old) were used and acquired from local farms authorized by SENASA (https://www.argentina.gob.ar/senasa, accessed on 15 April 2025; National Food Quality Service sanitation from the Argentinian government). The animals were kept individually, with a photoperiod of 12 h of light per day and a temperature of 18–25 °C. They were fed ad libitum with a commercial diet (GEPSA FEEDS^®^, Grupo Pilar SA, Trenque Lauquen, Buenos Aires, Argentina, 17% crude protein, 60.5% carbohydrates, 16% fiber, 0% saturated fats, 5.3% minerals, and 12% water).

At 6 months of age (adult), they were randomly distributed into 3 initial experimental groups (the experimental design is provided in Appendix A, whereas Table 1 shows the percentages of fat and olive oil supplementation for each rabbit group): the control diet (CD), fed with the commercial diet (normal diet, ND), the high-fat diet group (HFD), which received an experimental diet consisting in ND enriched with 14% *v*/*w* of cow fats, and the group fed with extra virgin olive oil (EVOO groups), with a diet enriched with 14% EVOO *v*/*w*. Cow fats correspond to commercial grease derived from the cow (“first bovine juice”: primer jugo bovino, https://www.argentina.gob.ar/anmat/codigoalimentario, accessed on 15 April 2025).

The CD and EVOO groups were maintained for 12 months on their respective diets for 12 months before being sacrificed. In the HFD group, after four months on the diet, the animals were divided into subgroups: one continued on the HFD, while the other switched to a diet with 50% reduced fat content (subgroup ½ HFD, 7% grease *v*/*w*). After an additional four months, the ½ HFD subgroup was further divided: one group remained on the ½ HFD, while the other received ½ HFD supplemented with EVOO at half the concentration used in the EVOO group (subgroup ½ FHD + ½ EVOO; 7% fat and 7% EVOO *v*/*w*). The HFD subgroup was sacrificed at 12 months, while the ½ HFD and ½ HFD + ½ EVOO subgroups (protective diet) were sacrificed after four months on their respective diets (Appendix A). During the 12 months of the study, a veterinarian regularly monitored the animals. The diets were prepared following the methods outlined in previous papers [8,22].

### 4.4. General Parameters

Biometrics parameters: every two weeks, weight was checked using a pediatric scale (Brand: Systel, Model: Vita). Other biometric measurements, such as nose-tail length (cm), neck circumference (cm), and abdomen circumference (cm), were also recorded using an inextensible metal measuring tape (with zero offsets to apply the crossover technique). The BMI of the animals was calculated from the weight (kg) and squared length (m^2^) values, providing a key indicator of the nutritional status and progression of the experimental model.

Biochemical parameters: monthly, 1 mL of blood was taken, selecting the marginal vein of the ear as the venipuncture point, after asepsis with 96° alcohol, in non-anesthetized animals. Heparinized syringes were used to prevent sample coagulation. The blood was centrifuged at 1100× *g* for 10 min. Determinations included: glucose, triglycerides, total cholesterol (TC), and HDL cholesterol (C-HDL). These analyses were performed with the GTLab kit (Rosario-Argentina), using the Trinder colorimetric enzymatic method, following the protocol provided by the manufacturer.

Seminal parameters: semen samples were collected and analyzed once a month. Artificial vaginas were used for this purpose. Semen samples were stored at 37 °C in thermostatic plates until analysis. Different parameters were assessed, including appearance, color, volume, and pH, the latter using MColorpHast pH indicator strips (Millipore, 109543). Sample viability was determined by staining with 0.5% eosin in PBS (eosin test for sperm vitality, WHO laboratory manual, third edition—1992). One drop of semen was mixed with one drop of eosin on a slide, then covered with a coverslip and observed under a light microscope. Unstained cells were considered alive, and the results were expressed as a percentage of the total sperm counted in 40 μL of semen [47]. Another aliquot of semen was used to assess concentration and motility. For this, the sample was diluted in PBS at 37 °C 1:50, (*v*/*v*), 20 μL were seeded in a Makler^®^ chamber (Counting Chamber, Sefi Medical Instruments, Haifa, Israel) and the number of spermatozoa was counted in 10 grids of the chamber. The number of spermatozoa counted was multiplied by the dilution factor and the final concentration expressed as 10^6^ spermatozoa/mL was obtained. In the same chamber, motility was evaluated by classifying the spermatozoa into three groups: progressively motile, which moved in a straight line; non-progressively motile, which showed movement without a specific direction; and immotile, which did not move. To study sperm morphology, a third aliquot of semen was washed three times in PBS, centrifuged for 10 min at 750× *g*, and finally, the resulting pellet was resuspended in a fixative solution (4% paraformaldehyde in PBS). A sperm smear was then performed, stained with Giemsa (Giemsa Stain, Modified Solution, No.: 51811-82-6), and evaluated under an optical microscope. The results of these studies have been previously published in several papers [8,9,21,22].

### 4.5. Structural Studies

Testis samples were immediately fixed after the animal was sacrificed, or kept in a buffer solution to isolate seminiferous tubules, see below.

Optical microscopy: sections of testis tissues obtained after sacrifice were fixed by immersing in a fixative solution composed of: 4% paraformaldehyde in PBS. They were then subjected to a progressive dehydration process using ethanol in increasing concentrations, starting at 50% and reaching 100%. Subsequently, the dehydrated sections were immersed in xylene and embedded in liquid paraffin. Once the paraffin had solidified at room temperature, 5 µm-thick sections were made using a sliding microtome. The sections were mounted on slides to be deparaffinized in xylene and rehydrated in ethanol–water solutions. Finally, they were stained with hematoxylin–eosin classical methods, and PAS (periodic acid stain), or immunostaining—see below.

Stage classification of seminiferous epithelium: rabbit seminiferous tubules cross sections stained with hematoxylin–eosin (H/E) and periodic acid–Schiff (PAS) were used. Images were then obtained with a Nikon 80i light microscope. ST stages were classified using the criteria described by Swierstra, 1963 [23]. These criteria include the shape of the spermatid nucleus, the location of spermatids and spermatozoa relative to the basement membrane, the presence of meiotic figures, and the release of spermatozoa into the lumen of the ST.

Seminiferous tubule isolation and characterization: the protocol described by Mäkelä et al. [24] was followed for isolation. The testes were removed from the rabbits, decapsulated, and the STs were left in a Petri dish containing PBS. They were then incubated for 10 min in a 0.5% collagenase solution in PBS at 37 °C. The dish was placed on a stereomicroscope (Zeiss, Oberkochen, Germany) with transmitted light that allowed the translucency of the STs to be visualized. The amount of light absorbed/scattered is directly related to the degree of chromatin condensation in the elongated spermatids and their clustering within the STs: as the chromatin becomes more condensed, there is greater light absorption, resulting in a darker appearance. The tubules were classified into three categories according to their clarity: clear (Zone 1), intermediate-light (Zone 2), and dark tubules (Zone 3). The tubule of interest was carefully lifted using hook-tipped forceps, and then a segment of the appropriate length was cut using microdissection scissors. These tubules were fixed and processed for optical microscopy or immune detection.

Cholesterol detection: filipin is an antifungal antibiotic naturally produced by the bacterium *Streptomyces filipinensis* [48,49]. Due to its high affinity for cholesterol, filipin is widely used as a selective marker for this lipid, also taking advantage of its autofluorescent property in the ultraviolet range [50,51]. This compound has been used as a histochemical marker for non-esterified cholesterol in numerous diseases, including Niemann–Pick Type C [52], Alzheimer’s disease [53], and Huntington’s disease [54]. For effective filipin staining, endogenous autofluorescence from tissues must be eliminated. The presence of lipofuscin (a fluorescent pigment accumulated in the cytoplasm) interferes with epifluorescence microscopy. For this reason, before each assay, tissue sections mounted on slides were exposed to neon white light (18 W) for 24 h followed by an additional 24 h under UV light (20 W), thereby significantly reducing the intensity of autofluorescence [52]. After this step, the tissues were deparaffinized by incubation in an oven at 60 °C for 1 h, followed by two washes in xylene for 15 min each. Then, the sections were rehydrated in alcohol solutions at decreasing concentrations (100, 96, 80, and 70%), for 5 min each, until reaching double-distilled water. Before treatment with filipin (Cayman Chemical, Ann Arbor, MI, USA, catalog no.: 70440), the sections were incubated for 15 min in PBS to equilibrate the medium. For cholesterol labeling, tissue sections were incubated with filipin in PBS for 2 h at room temperature, in a humid chamber, and in the dark. Sections were then washed 3 times in PBS and incubated with the propidium iodide nuclear marker (Sigma, P4170) at a 1/400 dilution for 30 min at room temperature and in a humid chamber. Finally, slides were washed 3 times with PBS and mounted with a Mowiol fluorescence medium (Sigma, 4-88). Confocal microscopy analysis was performed on an Olympus FV1000 microscope (Olympus America Inc., Center Valley, PA, USA). Five microscopic fields were randomly selected from each section to assess fluorescence intensity using ImageJ software (National Institutes of Health Bethesda, MD; https://imagej.net/ij/). The signal at 480 nm (blue channel) was tabulated as the mean (±SD) of five replicates per condition.

SREBP2 immunostaining: the immune location of SREBP2 was analyzed by indirect immunofluorescence. Before immune detection, autofluorescence was suppressed by exposing the slides to 24 h of white light and 24 h of UV light [50,51]. The samples were treated similarly to cholesterol detection, as described above. For epitope exposure, they were processed using sodium citrate buffer at 100 °C (in a water bath), for 30 min (0.01 M sodium citrate with 0.05% Tween-20 at pH 6). Then, Sudan black dye was used, which was applied for 25 min to reduce residual autofluorescence. The sections were washed 3 times in PBS for 5 min, under agitation. To block nonspecific binding sites, the sections were incubated in a humid chamber with blocking solution (1× PBS, 100× Triton, and 3% bovine serum albumin) for one hour, at room temperature. Double immunostaining was performed, starting with the incubation of the primary antibody anti-Arp 2/3 (Abcam, Cambridge, UK, ab115217), diluted in a blocking solution 1:100. The sections were incubated overnight at 4 °C in a humid chamber. The next day, the sections were washed with PBS three times for 5 min under agitation. The samples were then incubated with a biotinylated anti-rabbit secondary antibody (1:200) in an antibody buffer for 3 h at room temperature in a humid chamber, followed by three washes with PBS for 5 min under shaking. The samples were then incubated with the fluorophore streptavidin conjugated to Alexa 594 (1:200) for 2 h at room temperature. After three more washes with PBS (for 5 min, under shaking), the primary anti-alpha-tubulin antibody (MP, the primary anti-alpha-tubulin antibody, MP Biomedicals, 0869125), diluted 1:100 in blocking solution, was incubated overnight at 4 °C in a humid chamber. The following day, washes were performed with PBS (three times for 5 min under shaking) and the samples were incubated with an anti-mouse antibody labeled with Alexa Fluor 647 (1:200) in the antibody buffer solution for 2 h, at room temperature and in a humid chamber. During this incubation, the nuclear marker DAPI (4′,6-diamidino-2-phenylindole; Sigma, D9542) was also added at 1:500. After being washed three times with PBS under agitation and mounted with Mowiol (Sigma, 4-88), the sections were examined under an Olympus FV1000 confocal microscope (Olympus America Inc., Center Valley, PA, USA). The images obtained were analyzed with the ImageJ program. This protocol was also applied for isolated STs.

Fluorescence intensity quantification: for the analysis of SREBP2 and tubulin fluorescence signal intensity according to stage and diet, two images per tubule were taken with a confocal microscope (600×), selecting the tubules according to their stage, classified as clear (Z1), intermediate (Z2) and dark (Z3). This procedure was repeated in three animals of each group (n = 6 images/specimen). For each image, a projection on the Z axis of the set of optical planes was generated (plane thickness: 1 μm). Six quadrants with a surface of 273.5 μm^2^ were selected within each image (n = 6 quadrants/image) using the ROI Manager tool of the ImageJ software. Next, the mean fluorescence intensity (MFI) of the mark of interest was measured in each selected quadrant. Finally, the data were tabulated and analyzed by stages of the sperm cycle according to the Z1, Z2, and Z3 classification.

### 4.6. Molecular Studies

RNA extraction: the total ribonucleic acid (RNA) from the testis and liver was isolated using Trizol (Invitrogen, Waltham, MA, USA, 15596-026; 0.1 g of tissue was homogenized in 1 mL of Trizol solution). RNA was separated by adding 0.2 mL of chloroform and mixing for 15 s. Centrifugation was then performed at 12,000× *g* for 15 min at 4 °C. The upper aqueous phase was collected and transferred to a new tube. Isopropyl alcohol (0.5 mL per ml of Trizol) was added to the aqueous phase to precipitate the RNA. This RNA was washed twice with 75% ethanol and then centrifuged at 7500× *g* for 5 min at 4 °C. The pellet obtained was dried and resuspended in RNase-free water. RNA quality was determined by measuring the absorbance ratio at 260/280 nm wavelengths, complemented by visualization by electrophoretic running in 1% agarose gels.

mRNA expression analysis: mRNA expression was analyzed by reverse transcription (RT) followed by semi-quantitative polymerase chain reaction (PCR), i.e., relative to a constitutively expressed mRNA (actin). The isolated total RNA was reverse-transcribed to cDNA (copy deoxyribonucleic acid) with an Invitrogen kit. The reaction was performed in a thermocycler (brand: MPI, model: 01), incubating the mixture of RNA, deoxynucleotide triphosphates (dNTPs), and random primers in water for 5 min at 65 °C. It was then incubated for 2 min at 37 °C with 5 X buffer, DTT, a dithiothreitol-and-RNase-out enzyme –ribonuclease inhibitor. Finally, the mixture was incubated for 10 min at 25 °C, 50 min at 37 °C, and 15 min at 70 °C with the addition of the M-MLV reverse transcriptase enzyme. PCR amplified two μL of cDNA with 0.125 units of GoTaqDNA polymerase using selective primers designed for each case (Table 2), using the primer design program of the National Center for Biotechnology Information (NCBI), (https://www.ncbi.nlm.nih.gov/tools/primer-blast/, accessed 15 April 2025). The number of PCR cycles was adjusted according to the length of the mRNA of interest, optimizing the amplification of the cDNA without compromising the specificity or efficiency of the process. The PCR products were separated on 1% agarose gels and stained with SyBR (Invitrogen, S33102). Bands were visualized on a transilluminator (excitation: 280–502 nm, emission 530 nm), and images were obtained using the ImageQuant LAS 4000 system (GE Healthcare Bio-Sciences AB, Uppsala, Sweden). Densitometric analysis was performed with ImageJ software using the Analyze-Gels plugin. Expression levels of the genes of interest were normalized against actin expression levels, for which a species-specific primer was designed.

## 5. Conclusions

In this study, HFDs were shown to significantly alter cholesterol distribution and SREBP2 expression throughout the seminiferous epithelium, particularly during critical stages of spermatogenic differentiation, which correlates with a decline in sperm quality and functionality. The dysregulation of SREBP2—a central factor in maintaining testicular cholesterol homeostasis—creates a lipotoxic environment that compromises spermatogenesis, whereas supplementation with EVOO modulates SREBP2 activity and normalizes cholesterol distribution, thereby mitigating the adverse effects of a HFD and preserving the integrity of the spermatogenic process.

In this study, HFDs appeared to alter cholesterol distribution and SREBP2 expression throughout the seminiferous epithelium, particularly during the critical stages of spermatogenic differentiation. This may be associated with a decline in sperm quality and functionality. Although the limited sample size restricts the generalizability of the findings, the observed trends suggest that the dysregulation of SREBP2—a key factor in testicular cholesterol homeostasis—could contribute to a lipotoxic environment that compromises spermatogenesis. Interestingly, EVOO supplementation showed potential to modulate SREBP2 activity and normalize cholesterol distribution, thereby possibly mitigating the adverse effects of an HFD and helping to preserve spermatogenic integrity. Further studies with larger cohorts are needed to confirm these findings and elucidate the underlying mechanisms.

## Figures and Tables

**Figure 1 ijms-26-04062-f001:**
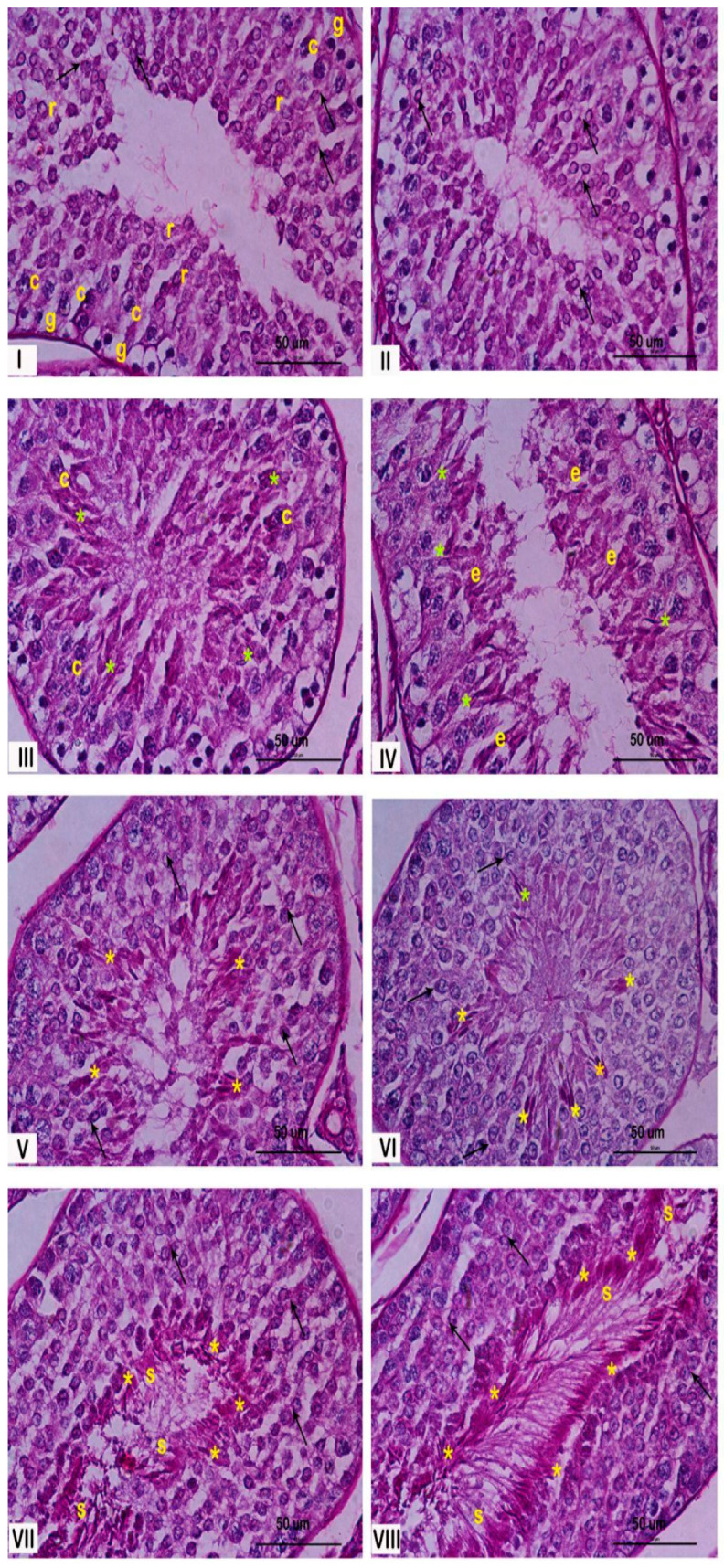
Histological characterization of the stages of the rabbit seminiferous epithelium. Representative cross sections of rabbit seminiferous tubules stained with PAS, illustrating the eight stages of the spermatogenic cycle, indicated by Roman numerals. **Stage I**: most of the spermatogenic cells could be recognized from spermatogonia (g), at the bottom of the epithelium, to round spermatids (r), close to the lumen of the seminiferous tubule. **Stage II**: the number of spermatogenic cells increase and begun to appear elongated spermatids (e). **Stage III**: the flagellum begun to develop and the lumen of the seminiferous tubules was blocked by them. **Stage IV**: elongated spermatids begun to aggregated and their development acrosome (*) were closely observed. **Stage V**: spermatids were closely distributed near the lumen. **Stage VI**: The spermatids are becoming increasingly thinner. The stratum of rounds spermatids was clearly distinguished. **Stage VII**: elongated spermatids were closed to the lumen and the residual body were easily detected. **Stage VIII**: the sperm cell was clearly delineated conformed the border of the lumen ready to support the spermiation process. The yellow asterisks (*) correspond to the acrosome in elongated spermatids and spermatozoa, black arrows indicate the acrosome in round spermatids at early stages, and letters denote specific cell types: r (round spermatids), c (spermatocytes), g (spermatogonia), e (elongated spermatids), and s (Sertoli cells). Magnification: 600×.

**Figure 2 ijms-26-04062-f002:**
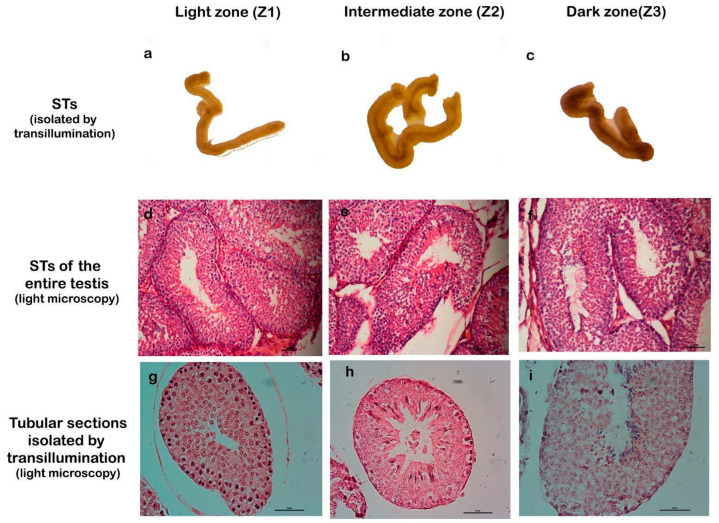
Correlation between transillumination zones and stages of the seminiferous epithelium. The first row (**a**–**c**) shows seminiferous tubules (STs) isolated by transillumination and classified into zones Z1, Z2, and Z3. The second row (**d**–**f**) shows STs of the entire testis, stained with H/E, which correspond to the proposed stages. The third row (**g**–**i**) corresponds to STs isolated by transillumination and processed for light microscopy, stained with H/E. Magnification: (**d**–**f**) 200×, (**g**–**i**) 400×.

**Figure 3 ijms-26-04062-f003:**
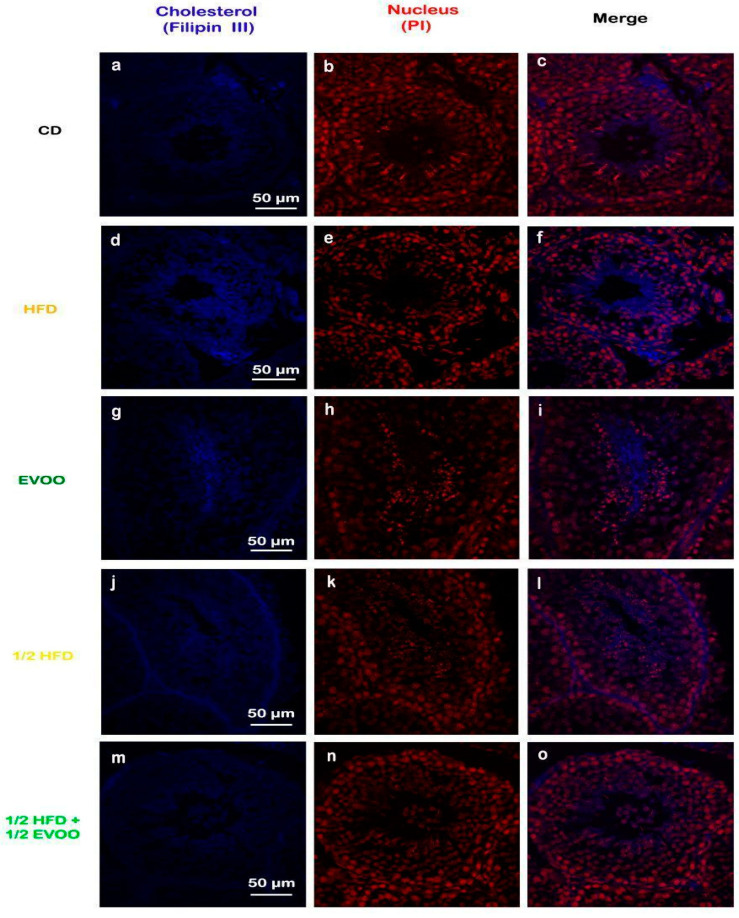
Cholesterol localization in the rabbit testis. Confocal microscopy images of testicular sections from adult rabbits stained with filipin (cholesterol, **a**,**d**,**g**,**j**,**m**), counterstained with propidium iodide (PI) to visualize nuclei (**b**,**e**,**h**,**k**,**n**), and merged (**c**,**f**,**i**,**l**,**o**) were presented in column, respectively. CD: rabbits on a normal diet (**a**,**b**,**c**); HFD: rabbits on a high-fat diet (**d**,**e**,**f**); EVOO: rabbits supplemented with extra virgin olive oil (**g**,**h**,**i**); ½ HFD: rabbits on a high-fat diet reduced by half (**j**,**k**,**l**); ½ HFD + ½ EVOO: rabbits fed a mixed diet (protective diet, **m**,**n**,**o**). Scale bar: 50 µm. Magnification: 600×.

**Figure 4 ijms-26-04062-f004:**
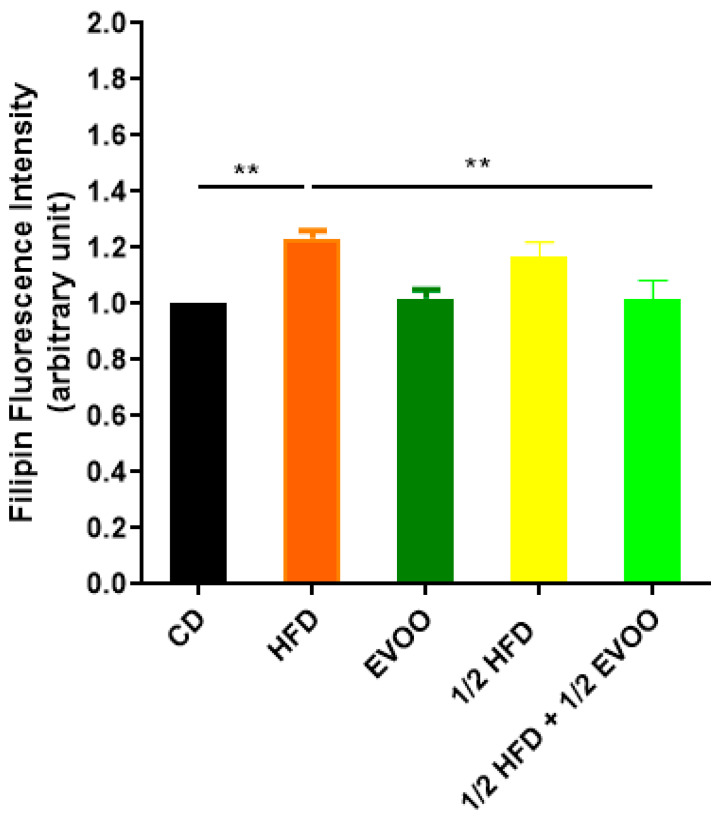
Quantification of cholesterol accumulation. Densitometry analysis of filipin staining. Results are expressed as the mean ± SD of the positive staining area, normalized to the control group (CD), (n = 4), and represented by colored bars. ** *p* < 0.01.

**Figure 5 ijms-26-04062-f005:**
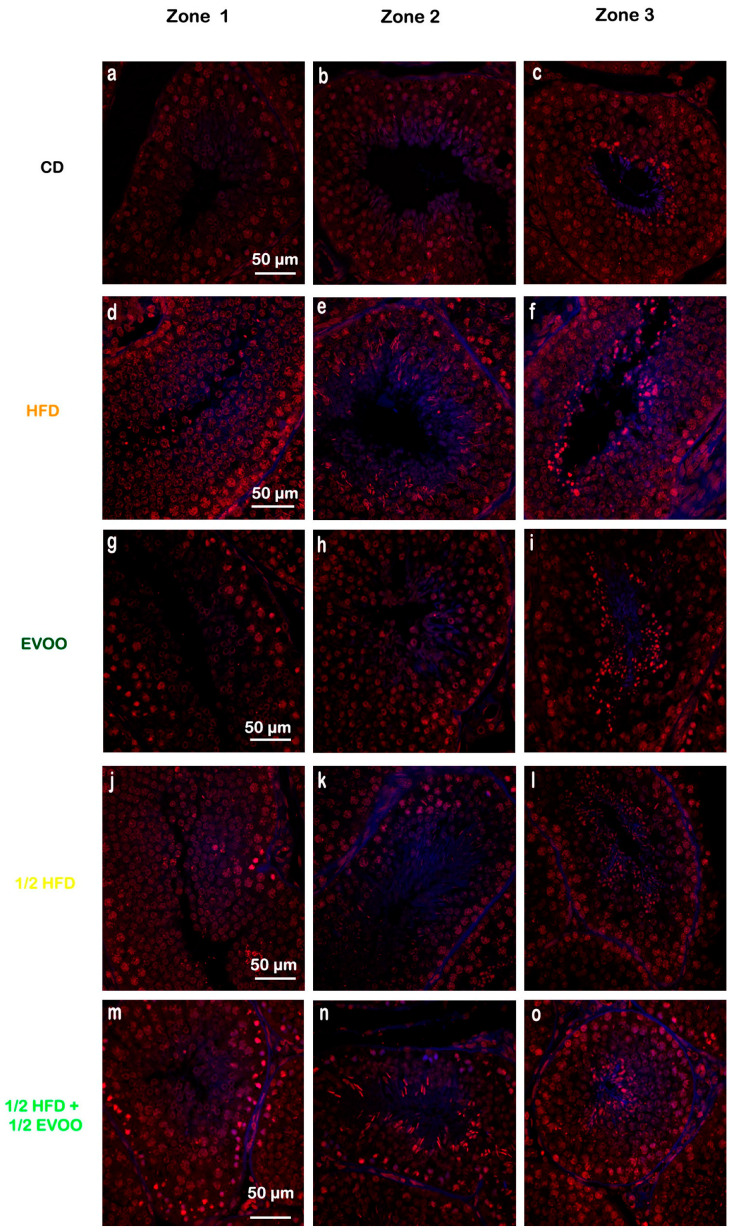
Cholesterol distribution in seminiferous tubules across different zones. Confocal micros-copy images of testicular sections from adult rabbits stained with filipin (cholesterol, blue fluorescence), and propidium iodide (PI, red fluorescence, cell nuclei). Isolated tubules were classified into three zones and presented in columns (left column: Zone 1, **a**,**d**,**g**,**j**,**m**; center column: Zone 2, **b**,**e**,**h**,**k**,**n**; and right column: Zone 3, **c**,**f**,**i**,**l**,**o**), based on transillumination: Zone 1 (I and II; clear tubules), Zone 2 (III, IV, V and VI; intermediate translucency), and Zone 3 (VII and VIII; dark tubules). Images of principal zones were also distributed in rows that correspond to diets: CD, rabbits on a normal diet (**a**,**b**,**c**); HFD, rabbits on a high-fat diet (**d**,**e**,**f**); EVOO, rabbits supplemented with extra virgin olive oil (**g**,**h**,**i**); ½ HFD, rabbits on a high-fat diet reduced by half (**j**,**k**,**l**); ½ HFD + ½ EVOO, rabbits fed a mixed diet (protective diet, **m**,**n**,**o**). Scale bar: 50 µm. Magnification: 600×.

**Figure 6 ijms-26-04062-f006:**
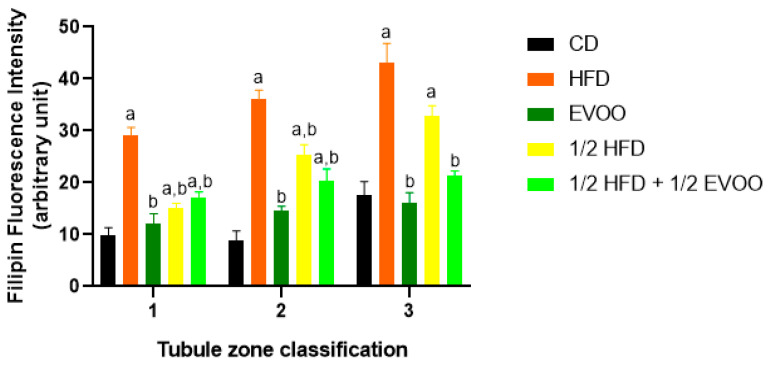
Quantification of filipin–cholesterol fluorescence intensity. The filipin–cholesterol fluorescence signal was quantified in each zone (*x*-axis, 1, 2, and 3) for each experimental group and represented by colored bars. Bars represent the mean ± SD. CD: rabbits on a normal diet (black bars); HFD: rabbits on a high-fat diet (orange bars); EVOO: rabbits supplemented with extra virgin olive oil (dark green bars); ½ HFD: rabbits on a high-fat diet reduced by half (yellow bars); ½ HFD + ½ EVOO: rabbits fed a mixed diet (protective diet, light green bars). Different letters indicate significant differences between groups within each zone (*p* < 0.05).

**Figure 7 ijms-26-04062-f007:**
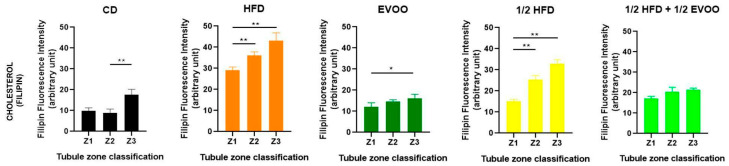
Quantification of cholesterol in different zones of the seminiferous tubule for each dietary group. Cholesterol distribution in zones Z1, Z2, and Z3 of the seminiferous tubule according to diet. Quantification of the mean fluorescence intensity (MFI) of filipin in different zones of the seminiferous tubule, represented as the mean ± SD (*n* = 4). CD: control diet; HFD: high-fat diet; EVOO: extra virgin olive oil; ½ HFD: half high-fat diet; ½ HFD + ½ EVOO, protective diet. Asterisks indicate significant differences between zones/stages of the spermatogenic cycle (* *p* < 0.05; ** *p* < 0.01). Multiple comparisons (Tukey test) corresponding to Filipin staining, according to the zone/stage of the seminiferous tubule epithelium, in each experimental group (Appendix A).

**Figure 8 ijms-26-04062-f008:**
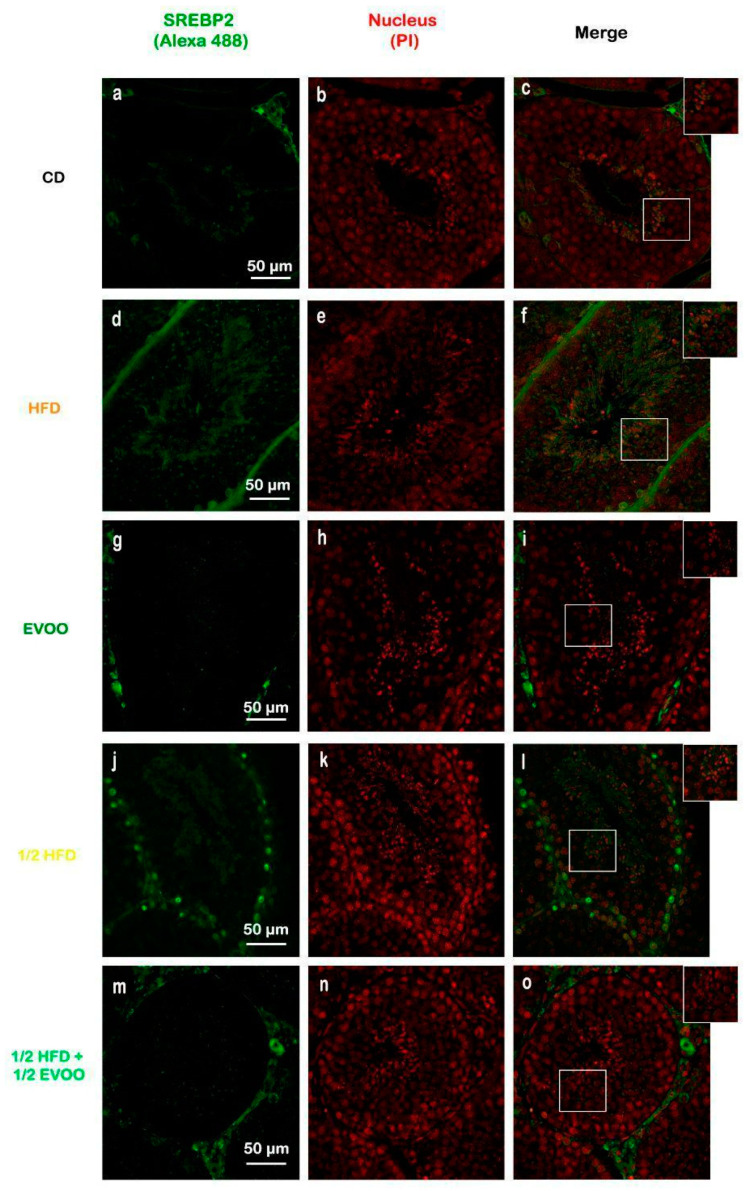
Immune location of SREBP2 in sections of seminiferous tubules corresponding to adult rabbit testes. Pictures was assemble using immune SREBP2 positive labels epithelial cells (Alexa 488, left column: **a**,**d**,**g**,**j**,**m**), propidium iodide (PI, cell nuclei, center column: **b**,**e**,**h**,**k**,**n**) and merge (right column, **c**,**f**,**i**,**l**,**o**) in columns and rows corresponding to diet (CD: rabbits on a normal diet, **a**,**b**,**c**; HFD: rabbits on a high-fat diet, **d**,**e**,**f**; EVOO: rabbits supplemented with extra virgin olive oil, **g**,**h**,**i**; ½ HFD: rabbits on a high-fat diet reduced by half, **j**,**k**,**l**; ½ HFD + ½ EVOO: rabbits fed a mixed diet—protective diet, **m**,**n**,**o**). White squares indicate the regions of interest used for densitometry analysis. Scale bar: 50 µm. Magnification: 600×.

**Figure 9 ijms-26-04062-f009:**
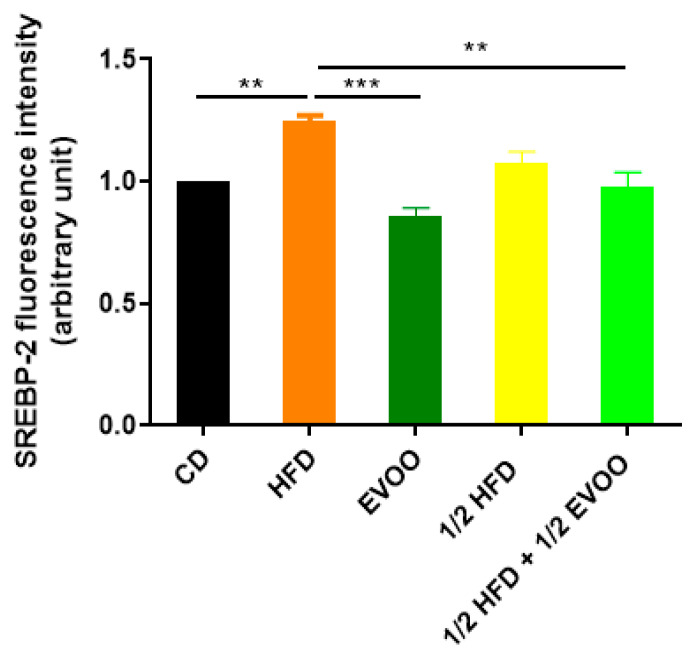
Quantification of SREBP2 fluorescence intensity. Densitometry analysis of the SREBP2 fluorescence intensity in the seminiferous tubules. Results are expressed as the mean ± SD, normalized to CD (1), (*n* = 4). ** *p* < 0.01; *** *p* < 0.001.

**Figure 10 ijms-26-04062-f010:**
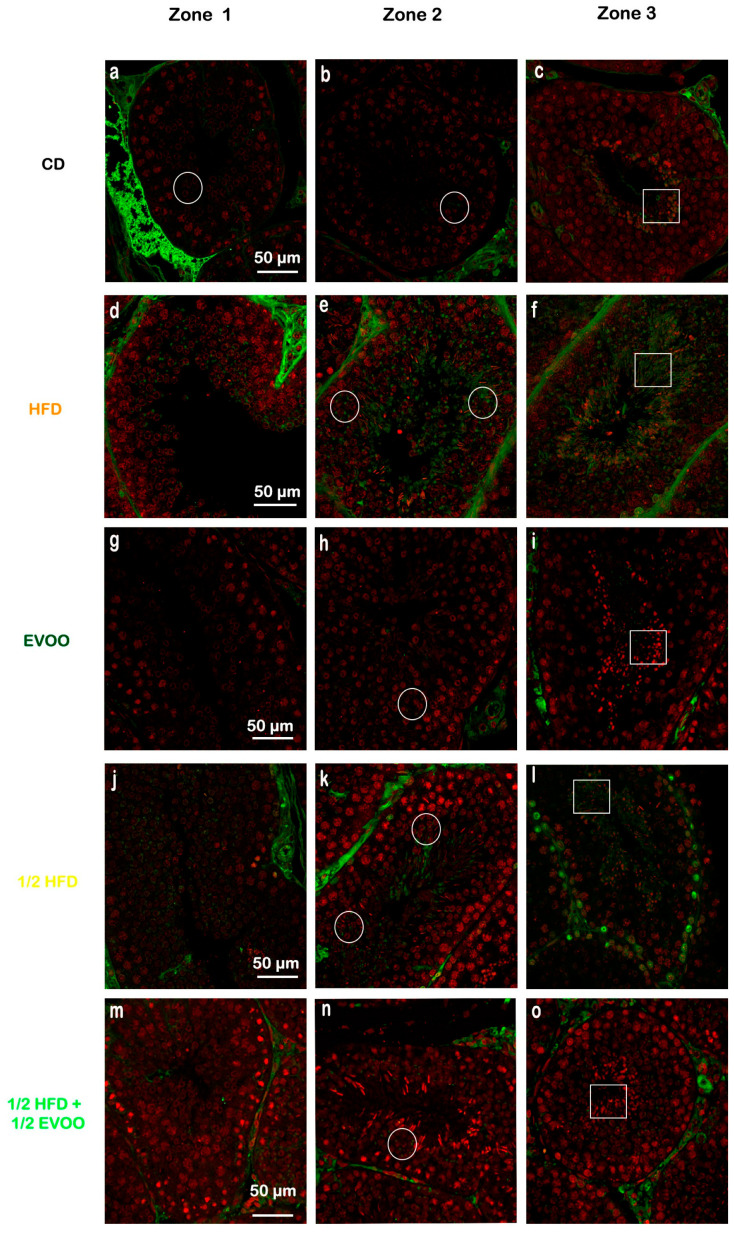
Localization of SREBP2 in the different zones/stages of seminiferous tubules under control and experimental diets. Representative immunofluorescence images of testicular sections from adult rabbits showing the localization of SREBP2 (in different zones of the seminiferous epithelium—columns—discriminated by type of experimental diets—rows). Zone 1 (Stage I and II; clear tubules, **a**,**d**,**g**,**j**,**m**), Zone 2 (Stage III, IV, V and VI; intermediate translucency, **b**,**e**,**h**,**k**,**n**) and Zone 3 (Stage VII and VIII; dark tubules, **c**,**f**,**i**,**l**,**o**). Cell nuclei were identified by propidium iodide (PI) staining. CD: rabbits on a normal diet (**a**,**b**,**c**); HFD: rabbits on a high-fat diet (**d**,**e**,**f**); EVOO: rabbits supplemented with extra virgin olive oil (**g**,**h**,**i**); ½ HFD: rabbits on a half-fat diet (**j**,**k**,**l**); ½ HFD + ½ EVOO: rabbits fed a mixed diet (protective diet, **m**,**n**,**o**). White circles and squares indicate regions of interest for densitometry analysis in the basal and apical compartments, respectively. Scale bar: 50 µm. Magnification: 600×.

**Figure 11 ijms-26-04062-f011:**
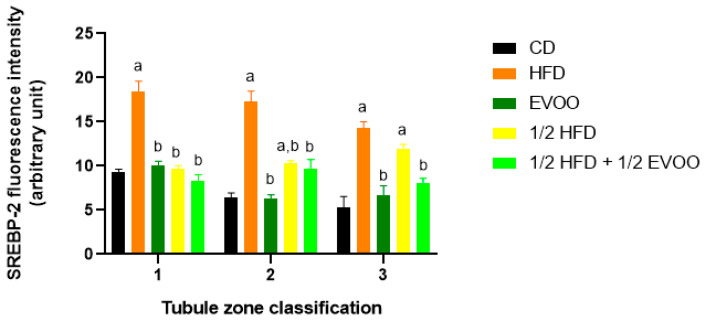
Quantification of SREBP2 fluorescence intensity in different zones of the seminiferous epithelium. The mean fluorescence intensity of SREBP2 in the basal and apical compartments of each zone (Z1, Z2, and Z3) for each dietary group. Data are presented as the mean ± SD (*n* = 4). Zone 1: stages I and II; Zone 2: stages III to VI; Zone 3: stages VII and VIII. CD: normal diet-fed rabbits; HFD: rabbits on the high-fat diet; AOVE: rabbits supplemented with extra virgin olive oil; ½ HFD: rabbits on grease reduced by half; ½ HFD + ½ AOVE: rabbits fed a mixed diet. Letters ‘a’ and ‘b’ indicate significant differences (*p* < 0.05) compared to the CD and HFD groups, respectively.

**Figure 12 ijms-26-04062-f012:**
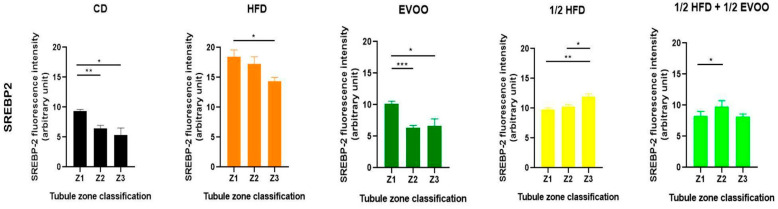
Distribution of SREBP2 among zones/stages of the seminiferous tubule according to diet. Quantification of SREBP2 fluorescence intensity in different zones of the seminiferous tubule. Data are presented as the mean ± SD (*n* = 4) for each dietary group: CD (control diet), HFD (high-fat diet), EVOO (extra virgin olive oil), ½ HFD (half high-fat diet), and ½ HFD + ½ EVOO (protected diet). Asterisks indicate significant differences between zones (Z1, Z2, and Z3) within each dietary group. Asterisks indicate significant differences between the zones/stages of the sperm cycle in each experimental group (Zone 1 = 1 (stages I and II); Zone 2 = 2 (stages III to VI); Zone 3 = 3 (stages VII and VIII)); * *p* < 0.05; ** *p* < 0.01; *** *p* < 0.001.

**Figure 13 ijms-26-04062-f013:**
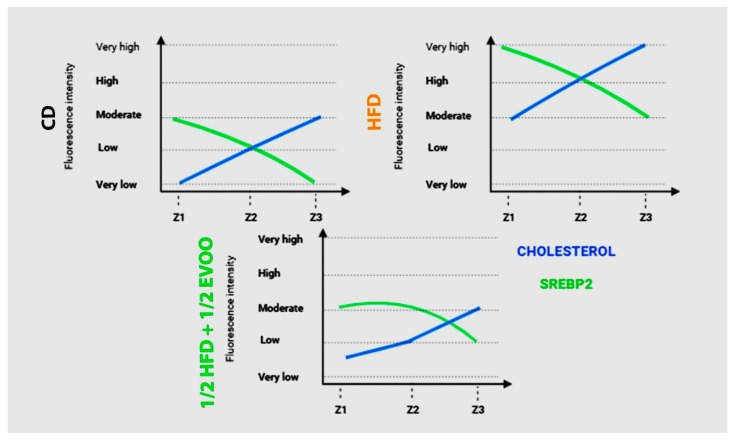
Comparison of changes in fluorescence intensity for SREBP2 and cholesterol across zones (stages of the seminiferous epithelium cycle) and diets. Fluorescence intensity is represented using arbitrary units, categorized into high, medium, and low expression levels. Data reflect differences between principal experimental diets (HFD, ½ HFD + ½ EVOO) and the control diet (CD).

**Figure 14 ijms-26-04062-f014:**
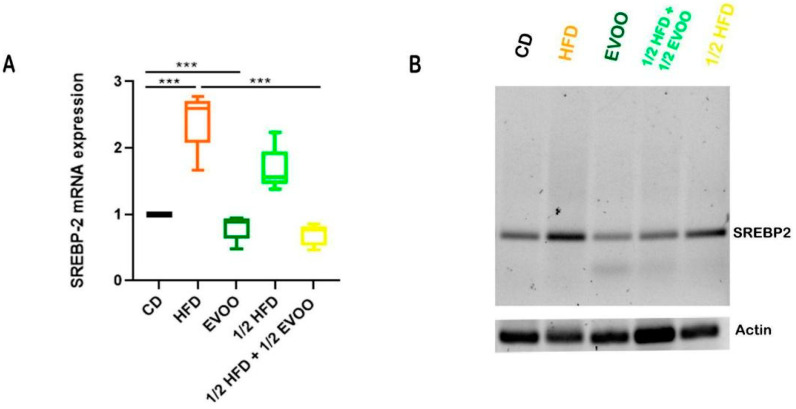
Expression of SREBP2 mRNA in rabbit testis under different diets. (**A**) Quantitative analysis of the relative expression of SREBP2 mRNA in testicular tissue from rabbits fed with different diets, corresponding to expression of SREBP2 mRNA—(**B**). Values represent the mean ± SD of three independent experiments, normalized to β-actin expression. CD: rabbits on a normal diet; HFD: rabbits on a high-fat diet; EVOO: rabbits supplemented with extra virgin olive oil; ½ HFD: rabbits on a high-fat diet reduced by half; ½ HFD + ½ EVOO: rabbits fed a mixed diet (protective diet). Asterisks indicate significant differences *p* < 0.001.

**Table 1 ijms-26-04062-t001:** Diets.

Diet	% (*v*/*w*) Fat Supplementation		% (*v*/*w*) Olive Oil Supplementation		Group Name
	14	7	14	7	
Normal	−	−	−	−	CD (control diet)
Experimental	+	−	−	−	HFD (high-fat diet)
	−	+	−	−	½ HFD (half high-fat diet)
	−	−	+	−	EVOO (extra virgin olive oil diet)
	−	+	−	+	½ HFD + ½ EVOO (half HFD and half extra virgin olive oil)—protective diet.

Percentages of fat and olive oil supplementation and categories of rabbits (groups) generated after feeding for 4 months. Normal diet (ND): commercial rabbit foods, and experimental diets: ND plus fat, extra virgin olive oil, or both (at 14% or 7% concentration *v*/*w*). An enrichment with 14 or 7% of the animal fat corresponds to HFD and ½ HFD, respectively; 14% of the olive oil corresponds to EVOO; and 7% fat + 7% EVOO corresponds to ½ HFD + ½ EVOO (protective diet). Note: +, indicates presence; and − indicates absence of the component.

**Table 2 ijms-26-04062-t002:** Sequence of primers used in PCR. T°: temperature at which the primers bind to the DNA strand. #C: number of cycles developed in the PCR reaction.

Primer	Forward	Reverse	T°	#C
Actin	ACCAACTGGGACGACATGGAGAA	GTCAGGATCTTCATGAGGTAGTC	54	30
SREBP2	CAGATTCCCTTGTTCTGACCACACTG	GCCAGCTTCAGCACCATGTTC	62	28

## Data Availability

Data are contained within the article and Appendix A.

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
