# Peer review of "Cholesterol and SREBP2 Dynamics During Spermatogenesis Stages in Rabbits: Effects of High-Fat Diet and Protective Role of Extra Virgin Olive Oil"

_ijms, 2025, doi:10.3390/ijms26094062_

Round 1
Reviewer 1 Report
Comments and Suggestions for Authors
The study presented by Avena et al. aims to determine how high-fat diets (HFDs) affect male fertility through cholesterol dysregulation and the role of the SREBP2 protein in this process. It also aims to evaluate the efficacy of extra virgin olive oil (EVOO) as a potential intervention to mitigate these effects and improve sperm quality.
To achieve this, the authors conducted research supported by various molecular techniques, yielding solid and conclusive results. They built on previous findings from their research team, representing an advancement in this field of knowledge. Their study demonstrates that SREBP2 expression is affected in rabbits fed high-fat diets. Currently, the molecular mechanisms by which these diets impair male fertility remain unknown, and this study may provide further insights into the issue. The sperm characteristics of rabbits fed such diets showed negative results in terms of quality and motility, among others. The authors assert that EVOO exerts a protective action by reducing cholesterol levels to those comparable to rabbits fed control diets.
However, to enhance the impact of this article, the following suggestions should be considered:
- There are unnecessary self-citations. While citing previous work by the authors is necessary, there appears to be an excessive number of self-citations. For example, references 2, 9, 11, 21, 22, and 23.
- A significant number of citations are from 1956 to 1998. It is recommended to replace some of these with more recent references.
- Figure 1 contains letters that are too small to distinguish clearly (e.g., r, c, g, and e).
- After line 108, it is recommended to indicate which figure is being described in the text.
- In line 540, the text refers to the experimental diets and indicates that Table 1 contains this information. However, Table 1 does not appear to reflect the general composition described in the text.
- In line 588, the information described in the text does not match the content of Table 2.
- Figure 9, as well as Tables 3 and 4, are not mentioned in the text.
- Some tables, such as Tables 1 and 2, could be moved to the supplementary materials section, as could Figure 7, since Figure 6 already provides all relevant information.
- Standard deviations are not clearly indicated in the graphs. These values should be provided in the supplementary materials.
- It is recommended to rewrite the conclusions section to be more specific and aligned with the results obtained. Conclusions 1 and 4 could be merged.
no comments
Author Response
Answer to reviewer 1
However, to enhance the impact of this article, the following suggestions should be considered:
- There are unnecessary self-citations. While citing previous work by the authors is necessary, there appears to be an excessive number of self-citations. For example, references 2, 9, 11, 21, 22, and 23.
We agree with the suggestions. Several self-citation was deleted, as well as some old ones: -original citation numbers 2, 9, 10, 11, 21, 22, and 24.
- A significant number of citations are from 1956 to 1998. It is recommended to replace some of these with more recent references.
Some references were included because they correspond to the original description of
specific topics. Some of them were deleted: please see the previous answer.
- Figure 1 contains letters that are too small to distinguish clearly (e.g., r, c, g, and e).
The letters were changed.
- After line 108, it is recommended to indicate which figure is being described in the text.
Thanks for the observations. we included the corresponding figure with the text.
- In line 540, the text refers to the experimental diets and indicates that Table 1 contains this information. However, Table 1 does not appear to reflect the general composition described in the text.
Many thanks for this observation. The experimental design describing the animal distribution and specific diets was removed to the supplementary material (Figure 1: experimental design). Table 1 contains the percentages of fat and olive oil supplementation by categories of rabbits (groups).
- In line 588, the information described in the text does not match the content of Table
Thanks for the observations. The table and the text were corrected to match adequately.
- Figure 9, as well as Tables 3 and 4, are not mentioned in the text.
-That's correct. The figure indication should have been included on lines 236-237. It was now changed to include the figure number.
-Line 236 – 237: Positive immune staining of SREBP2 indicates its presence and the amount of this regulatory molecule, under different diets in the seminiferous epithelium.
-New text: Positive immune staining of SREBP2 indicates its presence and the amount of this regulatory molecule, under different diets in the seminiferous epithelium (Figure 9).
- Some tables, such as Tables 1 and 2, could be moved to the supplementary materials section, as could Figure 7, since Figure 6 already provides all relevant information.
Yes, it is possible to move to supplementary materials.
- Standard deviations are not clearly indicated in the graphs. These values should be provided in the supplementary materials.
Many thanks for the suggestions. SD was included in the supplementary material.
- It is recommended to rewrite the conclusions section to be more specific and aligned with the results obtained. Conclusions 1 and 4 could be merged.
Thanks for the suggestions.
Conclusions were rewritten, merging points 1 a 4.
All text was revised - rewritten by an English-speaking person.
Reviewer 2 Report
Comments and Suggestions for Authors
There are some major corrections needed:
- Do not put fullstops in the end of manuscript title as well as chapter's titles
- There are many mistakes in the numbering of the tables and fogures: e.g. l 540 Table 1, General components of what? change the numeration of Tables - it is not clear. L 554 Table 1 should be Table 4??; l 565, l 588 and more - chceck all numbers of Tables and Figures
- EVOO has 70% of stearic acid and 9,9% of oleic acid?? Olive oil has mainly oleic acid...
- Check carefully the whole text.
I have no competence to asses the English language.
Author Response
Answer to Reviewer 2
- Do not put fullstops in the end of manuscript title as well as chapter's titles
Thanks for the suggestions. Full stops were deleted.
- There are many mistakes in the numbering of the tables and figures: e.g. l 540 Table 1, General components of what? change the numeration of Tables - it is not clear. L 554 Table 1 should be Table 4??; l 565, l 588 and more - check all numbers of Tables and Figures
Many thanks for the suggestions. The table number and content were adjusted
to be precise.
Figure 9 was included on page 10 – line 237
Tables were removed from the text to supplementary materials (Tables 1, 2, and 3 about olive oil quality).
Supplementary material also contains the figure of experimental design (Figure 1 in the supplementary materials)
- EVOO has 70% of stearic acid and 9,9% of oleic acid?? Olive oil has mainly oleic acid...
Many thanks for the comments. The table lost a line, and some numbers were incorrect. Please review the corrected table's supplementary material (Table 2).
- Check carefully the whole text.
An English-speaking person reviewed the text
Round 2
Reviewer 2 Report
Comments and Suggestions for Authors
Authors improved the manuscript according to reviewer's comments. However, the experimental design and not sufficient amount of animals (n=4) per group do not allow to clearly conclude the results. Update the conclusion section, please.
Comments on the Quality of English LanguageEnglish has been improved.
Author Response
Dear reviewer,
Thanks for your time and suggestion.
We have edited the conclusions; please see below.
I hope that the manuscript is adequate for IJMS.
Best regards
In this study, HFDs appeared to alter cholesterol distribution and SREBP2 expression throughout the seminiferous epithelium, particularly during critical stages of spermatogenic differentiation. This may be associated with a decline in sperm quality and functionality. Although the limited sample size restricts the generalizability of the findings, the observed trends suggest that the dysregulation of SREBP2—a key factor in testicular cholesterol homeostasis—could contribute to a lipotoxic environment that compromises spermatogenesis. Interestingly, EVOO supplementation showed a potential to modulate SREBP2 activity and normalize cholesterol distribution, thereby possibly mitigating the adverse effects of an HFD and helping to preserve spermatogenic integrity. Further studies with larger cohorts are needed to confirm these findings and elucidate the underlying mechanisms.